# Three-Way Fuzzy Sets and Their Applications (III)

**Qingqing Hu [1] and Xiaohong Zhang [2],\***

[1] School of Electrical and Control Engineering, Shaanxi University of Science & Technology, Xi'an 710021, China
[2] School of Mathematics & Data Science, Shaanxi University of Science & Technology, Xi'an 710021, China
**\*** Correspondence: zhangxiaohong@sust.edu.cn

**Abstract:** Three-way fuzzy inference is the theoretical basis of three-way fuzzy control. The proposed TCRI method is based on a Mamdani three-way fuzzy implication operator and uses one inference and simple composition operation. In order to effectively improve the TCRI method, this paper proposes a full implication triple I algorithm for three-way fuzzy inference and gives the triple I solution to the TFMP problem. The emphasis of our research is $R_0$ and Gödel triple I solution, which is related to three-way residual implication, as well as Zadeh's and Mamdani's triple I solution, which is based on three-way fuzzy implication operator. Then the three-way fuzzy controller is constructed by the proposed Zadeh's and $R_0$ triple I algorithm. Finally, the proposed triple I algorithm is applied to the three-way fuzzy control system, and its advantage is illustrated by the three-dimensional surface diagram of the control variable.

**Keywords:** three-way fuzzy sets; three-way fuzzy inference; three-way residual implication; Triple I Algorithm; three-way fuzzy control system

## 1. Introduction

Artificial intelligence is a new technical science that combines knowledge representation, knowledge acquisition, knowledge inference, knowledge driving with data learning, data mining, and data driving, and uses computers to simulate certain thinking processes and intelligent behaviors of humans. Considering the uncertainty of knowledge, various mathematical theories dealing with uncertainty have become the basis and key to artificial intelligence research. As the generalization of classical set theory, probability theory [1,2], fuzzy set theory [3] and rough set theory [4] have their own strengths in dealing with uncertain problems. In recent years, various generalized fuzzy set theories based on fuzzy set theory (such as intuitionistic fuzzy set [5–7], partial fuzzy set [8,9], neutrosophic set [10–13], picture fuzzy set [14] (known as totally dependent-neutrosophic sets [15])) and three-way decision theories based on rough set theory [16–19] have developed rapidly, and they play an important role in dealing with uncertainty problems in many fields such as information science, management science, and industrial process control. It is very meaningful to organically combine these theories with different characteristics and develop them into a mathematical tool that can effectively deal with uncertain problems, such as the combination of fuzzy sets and rough sets [20–22], and the combination of evidence theory and rough sets [23–25]. In 2022, Zhang et al. first proposed the concept of three-way fuzzy sets based on the basic ideas of three-way decision, rough sets and various generalized fuzzy sets, creating a new perspective of uncertainty theory [26]. The three-way fuzzy set skillfully combines the degree idea of fuzzy sets, generalized fuzzy sets and the idea of three-way decision, which makes it more extensive in describing the uncertainty of things (such as partial mapping instead of mapping), interpretability, and usability. Recently, Zhang et al. proposed rough set models based on three-way fuzzy sets, which are used in multi-criteria decision-making [27].

Fuzzy inference based on fuzzy sets is an important branch of uncertain inference, and it is the theoretical basis of fuzzy control systems, fuzzy neural network systems,

fuzzy decision support systems and so on. In 1973, Zadeh proposed the compositional rule of inference (abbreviated as CRI) to solve the fuzzy modus ponens (abbreviated as FMP) problem [28]. For more than 40 years, various fuzzy inference algorithms based on the CRI algorithm have occupied an important position in practical applications [29–34]. However, the CRI algorithm only uses one inference transformation. When the major premise is given to find the conclusion, it does not consider the relationship between the major premise and the conclusion, as well as the major premise and the minor premise. Instead, the relationship between the major premise and the minor premise is characterized by composition operations, which lacks complete inference meaning. Based on this, Wang proposed the full implication triple I algorithm of fuzzy inference in 1999 (referred to as the "Triple I Algorithm"), which provides a strict logical basis for fuzzy inference [35]. To date, the research on the triple I algorithm and its related theories has developed rapidly, and the results are very rich, mainly including the research on the triple I algorithm and its improved algorithms [36–42], the research on the related properties of triple I algorithm (reduction, robustness, constraint, support, approximation, etc.) [43–51], the research on fuzzy control system based on triple I algorithm [52–54] and so on.

Zhang et al. preliminarily studied three-way fuzzy inference, proposed the three-way compositional rule of inference (abbreviated as TCRI) based on the Mamdani three-way fuzzy implication operator, and discussed the application of this method in water level control system [26]. By calculating and comparing the dynamic performance indicators and immunity index quantification of the system, it is shown that the TCRI method has a better control effect and stronger anti-interference ability than the traditional fuzzy inference method. However, the logic foundation of three-way fuzzy inference is still weak, and the inference method is relatively simple, which needs to be further enriched and improved. In the traditional fuzzy inference methods, it is well known that the triple I algorithm with strict logic basis effectively improves the CRI method. Naturally, in order to effectively improve the TCRI method of three-way fuzzy inference, this paper puts forward the triple I algorithm of three-way fuzzy inference on the basis of three-way residual implication, discusses three-way fuzzy inference methods about several common three-way fuzzy implication operators, and achieves the process of inference with the help of computers, which lays a solid theoretical foundation for the application of three-way fuzzy inference in control systems.

Based on the basic theoretical research of three-way fuzzy sets, the rest of this paper is organized as follows. Section 2 shows several basic notions and an important theorem. In Section 3, three-way t-norm and its residual implication are introduced, $R_0$ and Gödel three-way adjoint pair are given. Furthermore, equivalent propositions related to the three-way adjoint pair are proved. Section 4 discusses triple I algorithms, which are based on $R_0$ and Gödel three-way residual implications, as well as Zadeh's and Mamdani's three-way fuzzy implication operators. Then, combined with the single input single output (abbreviated as SISO) problem and the double input single output (abbreviated as DISO) problem, the specific steps of $R_0$ and Zadeh's triple I solution are given. In addition, by comparing with TCRI inference results, the advantages of the proposed triple I algorithm are shown. In Section 5, the three-way fuzzy controller is designed and applied in control systems, and its superiority and effectiveness are described. Section 6 summarizes this paper and looks into future research.

## 2. Preliminaries

**Definition 1** ([26]). *Let $U$ be a non-empty set, $L$, $M$, $N$ be three lattices. The whole composed $\langle f, g, h \rangle$ of three partial mappings, which are defined as follows,*

$$f : U \to L; g : U \to M; h : U \to N.$$

*is called a three-way fuzzy set (abbreviated as TFS).*

Three-way fuzzy sets are usually represented by letters A, B, C, etc. A three-way fuzzy set $A$ on $U$ can be represented as follows,

$$A = \{\langle x, f_A(x), g_A(x), h_A(x) \rangle | x \in U, f_A(x) \in L, g_A(x) \in M, h_A(x) \in N\}.$$

According to the definition of three-way fuzzy set, there are some important related definitions as follows:

(1) If $L = M = N$, then $\langle f, g, h \rangle$ is called a lattice-valued three-way fuzzy set (or a $L$ three-way fuzzy set, or a regular three-way fuzzy set).

(2) Suppose that $\langle f, g, h \rangle$ is a three-way fuzzy set, $L = M = N = [0,1]$, and $\forall x \in U$, $f(x), g(x), h(x)$ are all defined, then $\langle f, g, h \rangle$ is called an ordinary three-way fuzzy set.

(3) Suppose that $\langle f, g, h \rangle$ is an ordinary three-way fuzzy set. If $f(x) + g(x) + h(x) \leq 1, \forall x \in U$, then $\langle f, g, h \rangle$ is called a dependent three-way fuzzy set; If $f(x) + g(x) + h(x) > 1, \forall x \in U$, then $\langle f, g, h \rangle$ is called an expanded three-way fuzzy set. For a dependent three-way fuzzy set $\langle f, g, h \rangle$, if $f(x), g(x), h(x) \in \{0,1\}, f(x) + g(x) + h(x) = 1, \forall x \in U$, then $\langle f, g, h \rangle$ is called a three-way set.

(4) Suppose that $\langle f, g, h \rangle$ is a regular three-way fuzzy set. If $f = g = h$, then $\langle f, g, h \rangle$ is called a one-way fuzzy set. A one-way fuzzy set is wider than a fuzzy set and a lattice-valued fuzzy set. If $g = h$, then $\langle f, g, h \rangle$ is called a two-way fuzzy set. A two-way fuzzy set is wider than an intuitionistic fuzzy set.

In order to further inspire, $TFS(U; L), TFS1(U), TFS2(U)$ are used to denote the distinct set consisting of $L$ three-way fuzzy sets, ordinary three-way fuzzy sets and dependent three-way fuzzy sets on $U$, respectively.

Suppose that $(L, \vee, \wedge, 0, 1)$ is a bounded lattice, where the order relation on $L$ is $\leq$. The binary relation $\leq_*$ on $L \cup \{*\}$ is defined as follows: $\forall x, y \in L \cup \{*\}$,

Case1: when $x, y \in L, x \leq y$, define $x \leq_* y$;

Case2: when $(x \in L - \{0,1\}, y = *)$ or $(y \in L - \{0,1\}, x = *)$, define $x \| y$ or $y \| x$;

Case3: when $x \in \{0, 1, *\}$, define $0 \leq_* * \leq_* 1$.

Obviously, $(L \cup \{*\}, \leq_*)$ is a bounded lattice. Furthermore, denote

$$D_*(L) = \{(x_1, x_2, x_3) \mid x_1, x_2, x_3 \in L \cup \{*\}\}.$$

If $x = (x_1, x_2, x_3) \in D_*(L)$, then $x_1, x_2, x_3$ are called the first, second, and third components of $x$, respectively.

**Definition 2** ([26]). *Suppose that $(L, \vee, \wedge, 0, 1)$ is a bounded lattice, where the order relation on $L$ is $\leq$. The binary relation $\leq_t$ on $D_*(L)$ is defined as follows: $\forall x, y \in D_*(L)$,*

$$x \leq_t y \Leftrightarrow (x_1 <_* y_1, x_3 \geq_* y_3) \text{ or } (x_1 = y_1, x_3 >_* y_3) \text{ or } (x_1 = y_1, x_3 = y_3, x_2 \leq_* y_2).$$

**Remark 1.** *The order relation of Definition 2 and its operation all have a precondition: $x_1$, $x_2$ and $x_3$ are three independent factors with clear positive, neutral and negative attributes. For example, the rationality of Example 1 and Example 2 in [27] is also judged according to actual problems.*

The related lattice structure of three-way fuzzy sets has been studied in [26] (Theorem 2), which is the theoretical basis for this paper to further study three-way t-norm and its residual implication.

**Theorem 1** ([26]). *Suppose that $(L, \vee, \wedge, 0, 1)$ is a bounded lattice, where the order relation on $L$ is $\leq$. Then, $(D_*(L), \leq_t)$ is a bounded partially ordered set, where $0_t = (0, 0, 1), 1_t = (1, 1, 0)$ are the minimal element and maximal element, respectively. Furthermore, $(D_*(L), \vee_t, \wedge_t, 0_t, 1_t)$ is a bounded lattice, $\forall x, y \in D_*(L)$,*

$$x \vee_t y = \begin{cases} x, & \text{if } y \leq_t x; \\ y, & \text{if } x \leq_t y; \\ (x_1 \vee y_1, 0, x_3 \wedge y_3) & \text{otherwise.} \end{cases}$$

$$x \wedge_t y = \begin{cases} x, & \text{if } x \leq_t y; \\ y, & \text{if } y \leq_t x; \\ (x_1 \wedge y_1, 1, x_3 \vee y_3) & \text{otherwise.} \end{cases}$$

**Definition 3** ([26]). *Suppose U is a universe, $TFS1(U)$ is the ordinary three-way fuzzy sets on U, $A \in TFS1(U)$. The complement of A (denote as $A^{ct}$) is defined as follows: $\forall x \in U$,*

$$A^{ct}(x) = (h_A(x), 1 - g_A(x), f_A(x)).$$

**Definition 4** ([55]). *Let $(L, \leq_L)$ be a complete lattice. A t-norm on $(L, \leq_L)$ is a commutative, associative, increasing mapping $T : L^2 \to L$, which satisfies $T(1_L, u) = u$, for all $u \in L$.*

According to the definitions of fuzzy implication and some common fuzzy implication operators [56], we proposed the definitions of three-way fuzzy implication operator and some common three-way fuzzy implication operators [26] as follows:

**Definition 5** ([26]). *A three-way fuzzy implication operator is a mapping "→": $D([0,1]) \times D([0,1]) \to D([0,1])$ that satisfies the following conditions:*
*(TFI1) $x \leq_t y \Rightarrow z \to x \leq_t z \to y$, $\forall x, y, z \in D([0,1])$;*
*(TFI2) $x \leq_t y \Rightarrow y \to z \leq_t x \to z$, $\forall x, y, z \in D([0,1])$;*
*(TFI3) $0_t \to 0_t = 1_t$;*
*(TFI4) $1_t \to 1_t = 1_t$;*
*(TFI5) $1_t \to 0_t = 0_t$.*

**Example 1** ([26]). *The definitions of common three-way fuzzy implication operators are as follows: $\forall x, y \in D([0,1])$,*
*(1) Mamdani's three-way fuzzy implication operator*

$$x \to_m y = x \wedge_t y.$$

*(2) Zadeh's three-way fuzzy implication operator*

$$x \to_z y = x^{ct} \vee_t (x \wedge_t y).$$

*(3) $R_0$ three-way fuzzy implication operator*

$$x \to_{r_0} y = \begin{cases} 1_t, & \text{if } x \leq_t y; \\ x^{ct} \vee_t y, & \text{otherwise.} \end{cases}$$

*(4) Gödel three-way fuzzy implication operator*

$$x \to_g y = \begin{cases} 1_t, & \text{if } x \leq_t y; \\ y, & \text{otherwise.} \end{cases}$$

**Remark 2.** *In traditional fuzzy set theory, Mamdani's implication is not a fuzzy implication operator in a strict sense, but it is often used in fuzzy control, so it is usually included in the list of common fuzzy implications, which belongs to abnormal implication (also known as non-boolean implication). Similarly, this method is also used here, but it should be pointed out that the Mamdani's operator does not fully meet the conditions in Definition 5. So, we use the Mamdani's three-way fuzzy implication operator in Example 1([23]) and this paper.*

## 3. Three-Way t-Norm and Its Residual Implication

$D_*(L)$ given by Theorem 1 is a bounded lattice, which is an important basis of this paper. So, we first give the proof of Theorem 1 as follows:

**Proof.** Firstly, we prove that $(D_*(L), \leq_t)$ is a partially ordered set.
　　(i) Reflexivity. By Definiton 2, there is obviously $x \leq_t x$, $\forall x \in D_*(L)$.
　　(ii) Ant-symmetry. Suppose $x, y \in D_*(L)$, $x \leq_t y$ and $y \leq_t x$.
　　From $x \leq_t y$, we get $(x_1 <_* y_1, x_3 \geq_* y_3)$ or $(x_1 = y_1, x_3 >_* y_3)$ or $(x_1 = y_1, x_3 = y_3, x_2 \leq_* y_2)$. Furthermore, from $y \leq_t x$, we get $(y_1 <_* x_1, y_3 \geq_* x_3)$ or $(y_1 = x_1, y_3 >_* x_3)$ or $(y_1 = x_1, y_3 = x_3, y_2 \leq_* x_2)$.
　　If $(x_1 <_* y_1, x_3 \geq_* y_3)$, then from $y \leq_t x$, we get $x_1 \geq_* y_1$, it is contradictory.

If $(x_1 = y_1, x_3 >_* y_3)$, then from $y \leq_t x$, we get $x_3 \leq_* y_3$, it is contradictory.

If $(x_1 = y_1, x_3 = y_3, x_2 \leq_* y_2)$, then from $y \leq_t x$, we get $y_2 \leq_* x_2$, so $x_2 = y_2$. Thus, $x_1 = y_1, x_2 = y_2, x_3 = y_3$, that is, $x = y$.

To sum up, if $x \leq_t y$ and $y \leq_t x$, then $x = y$.

(iii) Transitivity. Suppose $x, y, z \in D_*(L)$, $x \leq_t y$ and $y \leq_t z$.

(1) If $(x_1 <_* y_1, x_3 \geq_* y_3)$ and $(y_1 <_* z_1, y_3 \geq_* z_3)$, then $(x_1 <_* z_1, x_3 \geq_* z_3)$, that is, $x \leq_t z$.

(2) If $(x_1 <_* y_1, x_3 \geq_* y_3)$ and $(y_1 = z_1, y_3 >_* z_3)$, then $(x_1 <_* z_1, x_3 >_* z_3)$, that is, $x \leq_t z$.

(3) If $(x_1 <_* y_1, x_3 \geq_* y_3)$ and $(y_1 = z_1, y_3 = z_3, y_2 \leq_* z_2)$, then $(x_1 <_* z_1, x_3 \geq_* z_3)$, that is, $x \leq_t z$.

(4) If $(x_1 = y_1, x_3 >_* y_3)$ and $(y_1 <_* z_1, y_3 \geq_* z_3)$, then $(x_1 <_* z_1, x_3 >_* z_3)$, that is, $x \leq_t z$.

(5) If $(x_1 = y_1, x_3 >_* y_3)$ and $(y_1 = z_1, y_3 >_* z_3)$, then $(x_1 = z_1, x_3 >_* z_3)$, that is, $x \leq_t z$.

(6) If $(x_1 = y_1, x_3 >_* y_3)$ and $(y_1 = z_1, y_3 = z_3, y_2 \leq_* z_2)$, then $(x_1 = z_1, x_3 >_* z_3)$, that is, $x \leq_t z$.

(7) If $(x_1 = y_1, x_3 = y_3, x_2 \leq_* y_2)$ and $(y_1 <_* z_1, y_3 \geq_* z_3)$, then $(x_1 <_* z_1, x_3 \geq_* z_3)$, that is, $x \leq_t z$.

(8) If $(x_1 = y_1, x_3 = y_3, x_2 \leq_* y_2)$ and $(y_1 = z_1, y_3 >_* z_3)$, then $(x_1 = z_1, x_3 >_* z_3)$, that is, $x \leq_t z$.

(9) If $(x_1 = y_1, x_3 = y_3, x_2 \leq_* y_2)$ and $(y_1 = z_1, y_3 = z_3, y_2 \leq_* z_2)$, then $(x_1 = z_1, x_3 = z_3, x_2 \leq_* z_2)$, that is, $x \leq_t z$.

In summary, if $x \leq_t y$ and $y \leq_t z$, then $x \leq_t z$.

It is known from (i), (ii) and (iii) that $(D_*(L), \leq_t)$ is a partially ordered set.

Then, we prove that $(D_*(L), \vee_t, \wedge_t, 0_t, 1_t)$ is a bounded lattice.

(i) Suppose $x, y \in D_*(L)$, we prove that $x \vee_t y$ is the least upper bound of $x$ and $y$.

(1) If $x \leq_t y$ or $y \leq_t x$, it is obvious that $x \vee_t y$ is the least upper bound of $x$ and $y$.

(2) If $x \|_{\leq_t} y$, let $m = (m_1, m_2, m_3) = (x_1 \vee y_1, 0, x_3 \wedge y_3)$, then $x_1 \leq_* x_1 \vee y_1 = m_1$, $x_3 \geq_* x_3 \wedge y_3 = m_3$.

If $x_1 <_* m_1$ and $x_3 \geq_* m_3$, then $x \leq_t m$.

If $x_1 = m_1$ and $x_3 >_* m_3$, then $x \leq_t m$.

If $x_1 = m_1$ and $x_3 = m_3$, then $y_1 \leq_* x_1, y_3 \geq_* x_3$, that is, $y \leq_t x$ or $x \leq_t y$, it contradicts the condition $x \|_{\leq_t} y$.

In summary, $x \leq_t m$. Similarly, we can get $y \leq_t m$. Therefore, $m$ is the upper bound of $x, y$. Furthermore, we prove that $m$ is the least upper bound of $x, y$.

Assume $n = (n_1, n_2, n_3) \in D_*(L)$ such that $x \leq_t n, y \leq_t n$.

1) If $(x_1 <_* n_1, x_3 \geq_* n_3)$ and $(y_1 <_* n_1, y_3 \geq_* n_3)$, then $n_1 >_* x_1 \vee y_1 = m_1, n_3 \leq_* x_3 \wedge y_3 = m_3$, thus, $m \leq_t n$.

2) If $(x_1 <_* n_1, x_3 \geq_* n_3)$ and $(y_1 = n_1, y_3 >_* n_3)$, then $n_1 \geq_* x_1 \vee y_1 = m_1, n_3 \leq_* x_3 \wedge y_3 = m_3$. If $(n_1 >_* m_1, n_3 \leq_* m_3)$ or $(n_1 = m_1, n_3 <_* m_3)$, then $m \leq_t n$; If $n_1 = m_1$, $n_3 = m_3$, because $n_2 \geq_* 0$, so $m \leq_t n$.

3) If $(x_1 = n_1, x_3 >_* n_3)$ and $(y_1 <_* n_1, y_3 \geq_* n_3)$, then $n_1 \geq_* x_1 \vee y_1 = m_1, n_3 \leq_* x_3 \wedge y_3 = m_3$. Similar to 2), we can get $m \leq_t n$.

4) If $(x_1 = n_1, x_3 >_* n_3)$ and $(y_1 = n_1, y_3 >_* n_3)$, then $n_1 = x_1 \vee y_1 = m_1, n_3 <_* x_3 \wedge y_3 = m_3$, so $m \leq_t n$.

5) If $(x_1 = n_1, x_3 = n_3, x_2 \leq_* n_2)$ and $(y_1 = n_1, y_3 = n_3, y_2 \leq_* n_2)$, then $n_1 = x_1 \vee y_1 = m_1, n_3 = x_3 \wedge y_3 = m_3, n_2 \geq_* x_2 \vee y_2 = 0 = m_2$, so $m \leq_t n$.

6) If $(x_1 <_* n_1, x_3 \geq_* n_3)$ and $(y_1 = n_1, y_3 = n_3, y_2 \leq_* n_2)$, then $x_1 <_* y_1, x_3 \geq_* y_3$, so $x \leq_t y$, it contradicts the condition $x \|_{\leq_t} y$.

7) If $(x_1 = n_1, x_3 >_* n_3)$ and $(y_1 = n_1, y_3 = n_3, y_2 \leq_* n_2)$, then $x_1 = y_1, x_3 \geq_* y_3$, so $x \leq_t y$, it contradicts the condition $x \|_{\leq_t} y$.

8) If $(x_1 = n_1, x_3 = n_3, x_2 \leq_* n_2)$ and $(y_1 <_* n_1, y_3 \geq_* n_3)$, then $y_1 <_* x_1, y_3 \geq_* x_3$, so $y \leq_t x$, it contradicts the condition $x \|_{\leq_t} y$.

9) If $(x_1 = n_1, x_3 = n_3, x_2 \leq_* n_2)$ and $(y_1 = n_1, y_3 >_* n_3)$, then $y_1 = x_1, y_3 >_* x_3$, so $y \leq_t x$, it contradicts the condition $x\|_{\leq_t}y$.

In summary, $m \leq_t n$. Therefore, $m = (x_1 \vee y_1, 0, x_3 \wedge y_3)$ is th least upper bound of $x$ and $y$.

It is known from (1) and (2) that $x \vee_t y$ is the least upper bound of $x$ and $y$, for all $x, y \in D_*(L)$.

(ii) Suppose $x, y \in D_*(L)$, we prove that $x \wedge_t y$ is the greatest lower bound of $x$ and $y$.

(1) If $x \leq_t y$ or $y \leq_t x$, it is obvious that $x \wedge_t y$ is the greatest lower bound of $x$ and $y$.

(2) If $x\|_{\leq_t}y$, let $p = (p_1, p_2, p_3) = (x_1 \wedge y_1, 1, x_3 \vee y_3)$, then $x_1 \geq_* x_1 \wedge y_1 = p_1$, $x_3 \leq_* x_3 \vee y_3 = p_3$.

If $x_1 >_* p_1$ and $x_3 \leq_* p_3$, then $x \geq_t p$.

If $x_1 = p_1$ and $x_3 <_* p_3$, then $x \geq_t p$.

If $x_1 = p_1$ and $x_3 = p_3$, then $y_1 \geq_* x_1, y_3 \leq_* x_3$, that is, $y \geq_t x$ or $x \geq_t y$, it contradicts the condition $x\|_{\leq_t}y$.

In summary, $p \leq_t x$. Similarly, we can get $p \leq_t y$. Therefore, $p$ is the lower bound of $x, y$. Furthermore, we prove that $p$ is the greatest lower bound of $x, y$.

Assume $q = (q_1, q_2, q_3) \in D_*(L)$ such that $q \leq_t x, q \leq_t y$.

1) If $(q_1 <_* x_1, q_3 \geq_* x_3)$ and $(q_1 <_* y_1, q_3 \geq_* y_3)$, then $q_1 <_* x_1 \wedge y_1 = p_1, q_3 \geq_* x_3 \vee y_3 = p_3$, thus, $q \leq_t p$.

2) If $(q_1 <_* x_1, q_3 \geq_* x_3)$ and $(q_1 = y_1, q_3 >_* y_3)$, then $q_1 \leq_* x_1 \wedge y_1 = p_1, q_3 \geq_* x_3 \vee y_3 = p_3$. If $(q_1 <_* p_1, q_3 \geq_* p_3)$ or $(q_1 = p_1, q_3 >_* p_3)$, then $q \leq_t p$; If $q_1 = p_1, q_3 = p_3$, because $q_2 \leq_* 1 = p_2$, so $q \leq_t p$.

3) If $(q_1 = x_1, q_3 >_* x_3)$ and $(q_1 <_* y_1, q_3 \geq_* y_3)$, then $q_1 \leq_* x_1 \wedge y_1 = p_1, q_3 \geq_* x_3 \vee y_3 = p_3$. Similar to 2), we can get $q \leq_t p$.

4) If $(q_1 = x_1, q_3 >_* x_3)$ and $(q_1 = y_1, q_3 >_* y_3)$, then $q_1 = x_1 \wedge y_1 = p_1, q_3 >_* x_3 \vee y_3 = p_3$, so $q \leq_t p$.

5) If $(q_1 = x_1, q_3 = x_3, q_2 \leq_* x_2)$ and $(q_1 = y_1, q_3 = y_3, q_2 \leq_* y_2)$, then $q_1 = x_1 \wedge y_1 = p_1, q_3 = x_3 \vee y_3 = p_3, q_2 \leq_* x_2 \wedge y_2 = 1 = p_2$, so $q \leq_t p$.

6) If $(q_1 <_* x_1, q_3 \geq_* x_3)$ and $(q_1 = y_1, q_3 = y_3, q_2 \leq_* y_2)$, then $y_1 <_* x_1, y_3 \geq_* x_3$, so $y \leq_t x$, it contradicts the condition $x\|_{\leq_t}y$.

7) If $(q_1 = x_1, q_3 >_* x_3)$ and $(q_1 = y_1, q_3 = y_3, q_2 \leq_* y_2)$, then $x_1 = y_1, y_3 \geq_* x_3$, so $y \leq_t x$, it contradicts the condition $x\|_{\leq_t}y$.

8) If $(q_1 = x_1, q_3 = x_3, q_2 \leq_* x_2)$ and $(q_1 <_* y_1, q_3 \geq_* y_3)$, then $x_1 <_* y_1, x_3 \geq_* y_3$, so $x \leq_t y$, it contradicts the condition $x\|_{\leq_t}y$.

9) If $(q_1 = x_1, q_3 = x_3, q_2 \leq_* x_2)$ and $(q_1 = y_1, q_3 >_* y_3)$, then $x_1 = y_1, x_3 >_* y_3$, so $x \leq_t y$, it contradicts the condition $x\|_{\leq_t}y$.

In summary, $q \leq_t p$. Therefore, $p = (x_1 \wedge y_1, 1, x_3 \vee y_3)$ is the greatest lower bound of $x$ and $y$.

It is known from (1) and (2) that $x \wedge_t y$ is the greatest lower bound of $x$ and $y$, for all $x, y \in D_*(L)$.

Combing (i) and (ii), $(D_*(L), \leq_t)$ is a lattice.

Obviously, $1_t = (1, 1, 0)$, $0_t = (0, 0, 1)$ is the largest and smallest element of lattice $(D_*(L), \leq_t)$, respectively. Hence $(D_*(L), \vee_t, \wedge_t, 0_t, 1_t)$ is a bounded lattice. □

Suppose $D([0, 1]) = \{(x_1, x_2, x_3) \mid x_1, x_2, x_3 \in [0, 1]\}$. Obviously, $D([0, 1])$ is a special $D_*(L)$, then we can get the follwing theorem:

**Theorem 2.** *Suppose* $D([0, 1]) = \{(x_1, x_2, x_3) \mid x_1, x_2, x_3 \in [0, 1]\}$. *Then,* $(D([0, 1]), \vee_t, \wedge_t, 0_t, 1_t)$ *is a bounded lattice with inversely ordered involutive mapping.*

**Proof.** Obviously, $(D([0, 1]), \vee_t, \wedge_t, 0_t, 1_t)$ is a bounded lattice. Furthermore, for every $x \in D([0, 1])$, the mapping defined as follows:

$$h : D([0, 1]) \rightarrow D([0, 1]), h(x) = x^{ct}.$$

is an inversely ordered involutive mapping. □

In order to facilitate the study, the special bounded lattices $(D([0,1]), \vee_t, \wedge_t, 0_t, 1_t)$ is discussed in the rest of this paper. Furthermore, the notion of three-way t-norm and its residual implication on $D([0,1])$ are introduced. According to Theorem 1 and Theorem 2, it is obvious that $(D([0,1]), \leq_t)$ is a complete lattice. So, we expand the definition of t-norm [55] and get the definition of three-way t-norm on $(D([0,1]), \leq_t)$ as follows.

**Definition 6.** *A three-way t-norm is a function* $\otimes\colon (D([0,1]))^2 \to D([0,1])$ *such that for all* $x, y, z \in D([0,1])$:
*(TWT1)* $x \otimes y = y \otimes x$;
*(TWT2)* $(x \otimes y) \otimes z = x \otimes (y \otimes z)$;
*(TWT3)* $x \otimes y \leq_t x' \otimes y'$, *where* $x \leq_t x'$, $y \leq_t y'$;
*(TWT4)* $x \otimes 1_t = x$.

According to the definition of residual implication generated from a t-norm [56,57], we give the definition of the three-way residual implication generated from a three-way t-norm as follows:

**Definition 7.** *Suppose* $\otimes$ *is a three-way t-norm, then the three-way implication operator* $\to\colon$ $(D([0,1]))^2 \to D([0,1])$ *is called a three-way residual implication generated from a three-way t-norm* $\otimes$, *which is defined as follows:*

$$x \to y = \sup\{z | z \in D([0,1]), x \otimes z \leq_t y\}.$$

Furthermore, a three-way t-norm $\otimes$ satisfies the residual principle if and only if, for all $x, y, z \in D([0,1])$,

$$x \otimes z \leq_t y \Leftrightarrow z \leq_t x \to y. \tag{1}$$

When $\to$ is the three-way residual implication associated with a three-way t-norm $\otimes$, $(\otimes, \to)$ is called a three-way adjoint pair.

Several common three-way adjoint pairs on $D([0,1])$ are given as follows:

(1)   $R_0$ adjoint pairs $(\otimes_{r_0}, \to_{r_0})$

$$x \to_{r_0} y = \begin{cases} 1_t, & \text{if } x \leq_t y; \\ x^{ct} \vee_t y, & \text{otherwise.} \end{cases}$$

$$x \otimes_{r_0} y = \begin{cases} 0_t, & \text{if } x \leq_t y^{ct}; \\ x \wedge_t y, & \text{otherwise.} \end{cases}$$

(2)   Gödel adjoint pairs $(\otimes_g, \to_g)$

$$x \to_g y = \begin{cases} 1_t, & \text{if } x \leq_t y; \\ y, & \text{otherwise.} \end{cases}$$

$$x \otimes_g y = x \wedge_t y$$

**Remark 3.** *It is worth noting that the three-way fuzzy set on* $D([0,1])$ *is different from the neutrosophic set, mainly reflected in the following three aspects: (i) The three membership functions are independent of each other; (ii) Membership value can be missing; (iii) New inclusion relation and operations (the corresponding algebraic structure is generalized De Morgan algebra rather than De Morgan algebra).*

For the equivalence relationship between three-way t-norms and their residual implications, we give the following theorem according to [48].

**Theorem 3.** *Suppose that* $(\otimes, \to)$ *is the three-way adjoint pair on* $D([0,1])$. *Then*
*(1)* $1_t \otimes x = x$ *is equivalent to* $x \leq_t y \Leftrightarrow x \to y = 1_t$;
*(2)* $(x \otimes y) \otimes z = x \otimes (y \otimes z)$ *is equivalent to* $(x \otimes y) \to z = x \to (y \to z)$.

**Proof.** (1) Suppose that $1_t \otimes x = x$. Then $x \leq_t y \Leftrightarrow 1_t \otimes x \leq_t y \Leftrightarrow 1_t \leq_t x \to y \Leftrightarrow x \to y = 1_t$. Thus, $x \leq_t y \Leftrightarrow x \to y = 1_t$.

On the contrary, from $x \leq_t y \Leftrightarrow x \to y = 1_t$, we have $x \leq_t x \Leftrightarrow 1_t = x \to x$, then $1_t \leq_t x \to x \Leftrightarrow 1_t \otimes x \leq_t x$. Furthermore, from $1_t \otimes x \leq_t 1_t \otimes x$ and (1), we have $1_t \leq_t (x \to (1_t \otimes x))$, then $x \to (1_t \otimes x) = 1_t \Leftrightarrow x \leq_t 1_t \otimes x$. Thus, $1_t \otimes x = x$.

(2) Suppose that $(x \otimes y) \otimes z = x \otimes (y \otimes z)$. For $\forall m \in D([0,1])$, from $m \leq_t ((x \otimes y) \to z)$, we have $m \otimes (x \otimes y) \leq_t z \Leftrightarrow (m \otimes x) \otimes y \leq_t z \Leftrightarrow m \otimes x \leq_t (y \to z) \Leftrightarrow m \leq_t (x \to (y \to z))$. Thus, $(x \otimes y) \to z = x \to (y \to z)$.

On the contrary, for $\forall m \in D([0,1])$, from $(x \otimes y) \otimes z \leq_t m$, we have $x \otimes y \leq_t (z \to m) \Leftrightarrow x \leq_t (y \to (z \to m)) \Leftrightarrow x \leq_t ((y \otimes z) \to m) \Leftrightarrow x \otimes (y \otimes z) \leq_t m$. Thus, $(x \otimes y) \otimes z = x \otimes (y \otimes z)$. $\square$

## 4. The Triple I Algorithm of Three-Way Fuzzy Inference

Based on the triple I algorithm for FMP problem [35,48,58], we discussed the triple I algorithm of three-way fuzzy inference in this section.

The TFMP (Three-way Fuzzy Modus Ponens) Problem:

| Major premise: | $A \to B$ |
| Minor premise: | $A^*$ |
| --- | --- |
| Conclusion: | $B^*$ |

where $A, A^* \in TFS1(U)$, $B, B^* \in TFS1(V)$, $\to$ is the three-way fuzzy implication operator.

For the TFMP problem, Zhang et al. proposed the TCRI method, and the specific steps of this method are given [26]. On the basis of these works, the TCRI solutions based on Mamdani and Gödel three-way fuzzy implication operators are as follows:

$$B^*(v) = B(v) \wedge_t \left( \sup_{u \in U} \{ (A(u) \wedge_t A^*(u)) \} \right), \forall v \in V. \tag{2}$$

$$B^*(v) = \sup_{u \in U} \{ A^*(u) \wedge_t (A(u) \to_g B(v)) \}, \forall v \in V. \tag{3}$$

Although TCRI is successfully applied to fuzzy control problems [26], it only uses one inference transformation, which is, the major premise $A \to B$ is transformed into a three-way fuzzy relation on $U \times V$. However, when using the given minor premise $A^*$ to find $B^*$, the implication relation that should be satisfied between $A^* \to B^*$ and $A \to B$ is not considered. Wang proposed a triple I algorithm, which further improved the CRI method [35]. Motivated by Wang's triple I algorithm, we propose the triple I algorithm for the TFMP problem. The main idea of the triple I algorithm is that the minor premise $A^* \to B^*$ should be fully supposed by the major premise $A \to B$, which is, the solution $B^*$ to TFMP problem is the smallest three-way fuzzy set that maximizes the following formula,

$$(A(u) \to B(v)) \to (A^*(u) \to B^*(v)). \tag{4}$$

where $\to$ is the three-way fuzzy implication operator.

Then, the principle for the TFMP problem is given as follows:

Principle of Triple I Algorithm for TFMP problem: Suppose that $A, A^* \in TFS1(U)$, $B \in TFS1(V)$. Then the solution $B^*$ to the TFMP problem is the smallest three-way fuzzy set in $TFS1(V)$ which maximizes (4).

In particular, when the three-way fuzzy implication operator $\to$ satisfies the condition "$u \leq_t v \Leftrightarrow u \to v = 1_t$", $B^*$ is the smallest three-way fuzzy set that makes (4) always equal to $1_t$.

### 4.1. The Triple I Algorithm Based on Three-Way Residual Implication

Compared with different three-way fuzzy implication operators, we have different tripe I algorithms. The following theorem gives a unified triple I algorithm based on a three-way residual implication.

**Theorem 4.** *Suppose $\otimes$ is a three-way t-norm, $\to$ is the three-way residual implication associated with $\otimes$. Then the triple I solution to the TFMP problem exists uniquely and is given as follows:*

$$B^*(v) = \sup_{u \in U} \{A^*(u) \otimes (A(u) \to B(v))\}, \forall v \in V. \tag{5}$$

**Proof.** Firstly, for $u \in U$, $v \in V$, from the definition of $B^*$, we have $A^*(u) \otimes (A(u) \to B(v)) \leq_t B^*(v)$. According to (1), $A(u) \to B(v) \leq_t A^*(u) \to B^*(v)$, then $(A(u) \to B(v)) \to (A^*(u) \to B^*(v)) = 1_t$.

Next, let $C(v)$ be an arbitrary three-way fuzzy set on $V$ such that $(A(u) \to B(v)) \to (A^*(u) \to C(v)) = 1_t$, for $u \in U$, $v \in V$. Then $A(u) \to B(v) \leq_t A^*(u) \to C(v)$, according to (1), $A^*(u) \otimes (A(u) \to B(v)) \leq_t C(v)$, so $C(v)$ is the upper bound of $\{A^*(u) \otimes (A(u) \to B(v)) : u \in U\}$. Furthermore, from (5), $B^*(v)$ is the least upper bound of $\{A^*(u) \otimes (A(u) \to B(v)) : u \in U\}$, so, $B^*(v) \leq_t C(v)$, that is, $B^*$ is the smallest three-way fuzzy set.

Therefore, $B^*$ is the triple I solution to the TFMP problem. $\square$

**Corollary 1.** *Suppose that $\to$ in (4) is $R_0$ three-way residual implication. Then the triple I solution to (4) can be expressed as follows:*

$$B^*(v) = \sup_{u \in E_v} \{A^*(u) \wedge_t (A(u) \to_{r_0} B(v))\}, \forall v \in V. \tag{6}$$

*where*

$$E_v = \{u \in U \mid (A^*(u))^{ct} <_t (A(u) \to_{r_0} B(v))\}. \tag{7}$$

**Proof.** Since $\to$ in (4) is $R_0$ three-way residual implication, then (4) is written as follows:

$$(A(u) \to_{r_0} B(v)) \to_{r_0} (A^*(u) \to_{r_0} B^*(v)). \tag{8}$$

Assuming that $v, A, B$ and $A^*$ are all fixed, it is required to maximize the value of (8) for every $u$ and $B^*(v)$. Since $v$ is fixed, and let $D(u) = A(u) \to_{r_0} B(v)$, then (8) can be represented as follows:

$$D(u) \to_{r_0} (A^*(u) \to_{r_0} B^*(v)) = \begin{cases} 1_t, & \text{if } D(u) \leq_t (A^*(u) \to_{r_0} B^*(v)); \\ (D(u))^{ct} \vee_t (A^*(u) \to_{r_0} B^*(v)), & \text{otherwise.} \end{cases} \tag{9}$$

To make (9) achieve the maximum value of $1_t$, there must be,

$$D(u) \leq_t (A^*(u) \to_{r_0} B^*(v)) = \begin{cases} 1_t, & \text{if } A^*(u) \leq_t B^*(v); \\ (A^*(u))^{ct} \vee_t B^*(v), & \text{otherwise.} \end{cases} \tag{10}$$

If $A^*(u) \leq_t B^*(v)$, then (10) is always established.

If $A^*(u) >_t B^*(v)$, then to make (10) hold, there must be $(A^*(u))^{ct} \vee_t B^*(v) \geq_t D(u)$. If $(A^*(u))^{ct} \geq_t D(u)$, then it is nothing to do with $B^*(v)$; If $(A^*(u))^{ct} <_t D(u)$, then $B^*(v) \geq_t D(u)$.

To sum up,

$$B^*(v) = \sup_{u \in E_v} \{A^*(u) \wedge_t (A(u) \to_{r_0} B(v))\}, \forall v \in V,$$

where $E_v = \{u \in U \mid (A^*(u))^{ct} <_t (A(u) \to_{r_0} B(v))\}$. $\square$

**Corollary 2.** *Suppose that $\to$ in (4) is Gödel three-way residual implication. Then the triple I solution to (4) can be expressed as follows:*

$$B^*(v) = \sup_{u \in U} \{A^*(u) \wedge_t (A(u) \to_g B(v))\}, \forall v \in V. \tag{11}$$

**Proof.** Obviously, in (5), if $\otimes = \otimes_g = \wedge_t$ and $\to = \to_g$, then (5) is (11). □

**Remark 4.** *The triple I solution to TFMP problem given in Corollary 2 based on Gödel three-way residual implication is exactly the same as (3).*

*4.2. The Triple I Algorithm Based on Three-Way Fuzzy Implication Operator*

Fuzzy inference methods based on Zadeh's and Mamdani's implication operators are also widely used in fuzzy control. Therefore, we discuss triple I algorithms with regard to these two three-way fuzzy implication operators.

**Theorem 5.** *Suppose that $\to$ in (4) is Zadeh's three-way fuzzy implication operator. Then the triple I solution to (4) can be expressed as follows:*

$$B^*(v) = \sup_{u \in E_v, u \in Q_v} \{A^*(u) \wedge_t (A(u) \to_z B(v))\}, \forall v \in V. \tag{12}$$

*where*

$$\begin{aligned} E_v &= \{u \in U \mid (A^*(u))^{ct} <_t (A(u) \to_z B(v))\} \\ Q_v &= \{u \in U \mid (f_{(A(u) \to_z B(v))} = h_{(A(u) \to_z B(v))}, g_{(A(u) \to_z B(v))} > 0.5) \\ &\quad or f_{(A(u) \to_z B(v))} > h_{(A(u) \to_z B(v))}\}. \end{aligned} \tag{13}$$

**Proof.** Since $\to$ in (4) is Zadeh's three-way fuzzy implication operator, then (4) is written as follows:

$$(A(u) \to_z B(v)) \to_z (A^*(u) \to_z B^*(v)). \tag{14}$$

According to the definition of Zadeh's three-way fuzzy implication operator in Example 1, (14) is equivalent to

$$(A(u) \to_z B(v))^{ct} \vee_t ((A(u) \to_z B(v)) \wedge_t (A^*(u) \to_z B^*(v))). \tag{15}$$

Assuming that $v, A, B$ and $A^*$ are all fixed, it is required to maximize the value of (15) for every $u$ and $B^*(v)$. Since $v$ is fixed, and let $D(u) = A(u) \to_z B(v)$, then (15) can be represented as follows:

$$(D(u))^{ct} \vee_t (D(u) \wedge_t (A^*(u) \to_z B^*(v))). \tag{16}$$

If $(D(u))^{ct} \geq_t D(u)$, that is, $(f_D(u) = h_D(u), g_D(u) \leq 0.5)$ or $f_D(u) < h_D(u)$, then (16) can be transformed into $(D(u))^{ct}$, so, the maximum value of (16) has nothing to do with $B^*(v)$.

If $(D(u))^{ct} <_t D(u)$, that is, $(f_D(u) = h_D(u), g_D(u) > 0.5)$ or $f_D(u) > h_D(u)$, then (16) can be be transformed into

$$D(u) \wedge_t ((D(u))^{ct} \vee_t (A^*(u) \to_z B^*(v))). \tag{17}$$

To make (17) obtain the maximum value $D(u)$, there must be

$$A^*(u) \to_z B^*(v) = (A^*(u))^{ct} \vee_t (A^*(u) \wedge_t B^*(v)) \geq_t D(u). \tag{18}$$

If $(A^*(u))^{ct} \geq_t D(u)$, then the establishment of (18) has nothing to do with $B^*(v)$.

If $(A^*(u))^{ct} <_t D(u)$, then

$$A^*(u) \wedge_t B^*(v) \geq_t D(u). \tag{19}$$

If $A^*(u) <_t D(u)$, then (19) does not hold. So, let $B^*(v) \geq_t A^*(u)$, it can be written as $B^*(v) \geq_t A^*(u) \wedge_t D(u)$. If $A^*(u) \geq_t D(u)$, there must be that $B^*(v) \geq_t D(u)$, it can also be written as $B^*(v) \geq_t A^*(u) \wedge_t D(u)$.

To sum up,

$$B^*(v) = \sup_{u \in E_v, u \in Q_v} \{A^*(u) \wedge_t (A(u) \to_z B(v))\}, \forall v \in V.,$$

where $E_v = \{u \in U \mid (A^*(u))^{ct} <_t (A(u) \to_z B(v))\}$, $Q_v = \{u \in U \mid (f_{(A(u) \to_z B(v))} = h_{(A(u) \to_z B(v))}, g_{(A(u) \to_z B(v))} > 0.5)$ or $f_{(A(u) \to_z B(v))} > h_{(A(u) \to_z B(v))}\}$.  □

**Theorem 6.** *Suppose that $\to$ in (4) is a Mamdani's three-way fuzzy implication operator. Then the triple I solution to (4) can be expressed as follows:*

$$B^*(v) = B(v) \wedge_t (\sup_{u \in U} \{(A(u) \wedge_t A^*(u))\}), \forall v \in V. \tag{20}$$

**Proof.** Since $\to$ in (4) is Mamdani's three-way fuzzy implication operator, then (4) is written as follows:

$$(A(u) \to_m B(v)) \to_m (A^*(u) \to_m B^*(v)). \tag{21}$$

Assuming that $v, A, B$ and $A^*$ are all fixed, it is required to maximize the value of (21) for every $u$ and $B^*(v)$. Since $v$ is fixed, and let $D(u) = A(u) \to_m B(v)$, then (21) can be represented as follows:

$$D(u) \to_m (A^*(u) \to_m B^*(v)) = D(u) \wedge_t (A^*(u) \to_m B^*(v)). \tag{22}$$

Then (22) achieve the maximum value of $D(u)$, when

$$A^*(u) \to_m B^*(v) = A^*(u) \wedge_t B^*(v) \geq_t D(u), \tag{23}$$

If $A^*(u) <_t D(u)$, then (23) is not established. Let $B^*(v) \geq_t A^*(u)$, it can also be written as $B^*(v) \geq_t A^*(u) \wedge_t D(u)$.

If $A^*(u) \geq_t D(u)$, there must be $B^*(v) \geq_t D(u)$, it can be written as $B^*(v) \geq_t A^*(u) \wedge_t D(u)$.

To sum up,

$$B^*(v) = B(v) \wedge_t (\sup_{u \in U} \{(A(u) \wedge_t A^*(u))\}), \forall v \in V.$$

□

**Remark 5.** *The triple I solution to the TFMP problem given in Theorem 6 based on Mamdani's three-way fuzzy implication operator is exactly the same as (2).*

*4.3. Three-Way Fuzzy Inference Based on Triple I Algorithm*

Based on Sections 4.1 and 4.2, we focus on the specific steps of the triple I algorithm based on Zadeh's three-way implication operator and $R_0$ three-way residual implication with the following examples.

**Example 2.** *Let $U = V = \{1, 2, 3\}$, $A = \langle(0.8, 0.5, 0.2), (0.4, 0.6, 0.2), (0.1, 0.5, 0.9)\rangle$, $A^* = \langle(0.9, 0.2, 0.1), (0.3, 0.5, 0.3), (0.2, 0.7, 0.8)\rangle$ are three-way fuzzy sets on U, $B = \langle(0.2, 0.7, 0.9), (0.6, 0.5, 0.5), (0.7, 0.4, 0.1)\rangle$ is a three-way fuzzy set on V. Find $B^*$ according to the TCRI method and the triple I algorithm, which are based on Zadeh's three-way fuzzy implication operator.*

Suppose that $\rightarrow$ is Zadeh's three-way fuzzy implication operator $\rightarrow_z$. The fuzzy matrix $R$ is used to represent the three-way fuzzy relation as follows:

$$R = (r_{ij})_{3 \times 3} = (A(u_i) \rightarrow_z B(v_j))_{3 \times 3} = ((A(u_i))^{ct} \vee_t (A(u_i) \wedge_t B(v_j)))_{3 \times 3}.$$

$$R = \begin{bmatrix} (0.2, 0.5, 0.8) & (0.6, 0.5, 0.5) & (0.7, 1.0, 0.2) \\ (0.2, 0.4, 0.4) & (0.4, 0.0, 0.4) & (0.4, 0.6, 0.2) \\ (0.9, 0.5, 0.1) & (0.9, 0.5, 0.1) & (0.9, 0.5, 0.1) \end{bmatrix}.$$

(1)   $B^*$ is calculated using the TCRI method.

$$\begin{aligned} B^* &= A^* \circ (A \rightarrow_z B) \\ &= A^* \circ R \\ &= \langle (0.2, 0.4, 0.4), (0.6, 0.0, 0.4), (0.7, 1.0, 0.2) \rangle. \end{aligned} \tag{24}$$

(2)   $B^*$ is calculated using the triple I algorithm. From $(A^*)^{ct} <_t R$ and $R^{ct} <_t R$, we have

$$B^* = \langle (0.2, 0.7, 0.8), (0.6, 0.5, 0.5), (0.7, 1.0, 0.2) \rangle. \tag{25}$$

Comparing (24) with (25), it can be seen that $B^*$ obtained by the triple I algorithm is less than or equal to $B^*$ obtained by the TCRI method. Therefore, in terms of the meaning of seeking the smallest $B^*$ on $V$, the triple I algorithm is better than the TCRI method.

**Example 3.** *Let the universes $U$ and $V$, three-way fuzzy sets $A$, $B$, and $A^*$ be the same as those in Example 2. Find $B^*$ according to the TCRI method and the triple I algorithm, which are based on $R_0$ three-way residual implication.*

Suppose that $\rightarrow$ is $R_0$ three-way residual implication $\rightarrow_{r_0}$. The fuzzy matrix $R$ is used to represent the three-way fuzzy relation as follows:

$$R = \begin{bmatrix} (0.2, 0.5, 0.8) & (0.6, 0.5, 0.5) & (0.7, 0.4, 0.1) \\ (0.2, 0.4, 0.4) & (0.6, 0.0, 0.4) & (1.0, 1.0, 0.0) \\ (1.0, 1.0, 0.0) & (1.0, 1.0, 0.0) & (1.0, 1.0, 0.0) \end{bmatrix}.$$

(1)   $B^*$ is calculated using the TCRI method.

$$\begin{aligned} B^* &= A^* \circ (A \rightarrow_{r_0} B) \\ &= \langle (0.2, 0.4, 0.4), (0.6, 0.0, 0.4), (0.7, 0.4, 0.1) \rangle. \end{aligned} \tag{26}$$

(2)   $B^*$ is calculated using the triple I algorithm. From $(A^*)^{ct} <_t R$, we have

$$B^* = \langle (0.2, 0.7, 0.8), (0.6, 0.5, 0.5), (0.7, 0.4, 0.1) \rangle. \tag{27}$$

According to (26) and (27), since $0.2 = 0.2$, $0.8 > 0.4$, $(0.2, 0.7, 0.8) \leq_t (0.2, 0.4, 0.4)$; since $0.6 = 0.6$, $0.5 > 0.4$, $(0.6, 0.5, 0.5) \leq_t (0.6, 0.0, 0.4)$; $(0.7, 0.4, 0.1) = (0.7, 0.4, 0.1)$. Thus, $\langle (0.2, 0.7, 0.8), (0.6, 0.5, 0.5), (0.7, 0.4, 0.1) \rangle \leq_t \langle (0.2, 0.4, 0.4), (0.6, 0.0, 0.4), (0.7, 0.4, 0.1) \rangle$. So, the result of the triple I algorithm is smaller than the TCRI method.

It can be seen from the above two examples:

(1)   For the same sets of given: $A$, $B$ and $A^*$, the results are different because of the different $R$, which are generated by different three-way fuzzy implication operators.

(2)   For the same sets of given: $A$, $B$ and $A^*$, and the same three-way fuzzy implication operator, the results are different due to different inference methods, and the triple I algorithm is better.

In practical fuzzy control systems, there are usually many fuzzy rules, so the multi-rule three-way fuzzy inference based on the triple I algorithm is studied in the following section.

Combined with the two DISO examples, the general steps of three-way fuzzy inference based on Zadeh's three-way implication operator and $R_0$ three-way residual implication are described.

**Example 4.** *Let $U = \{u_1, u_2, u_3\}$, $V = \{v_1, v_2, v_3\}$ and $W = \{w_1, w_2, w_3\}$. Suppose that there is a three-way fuzzy control system with dual input (u and v) and single output (w). The two three-way fuzzy rules are as follows:*

$$R_1\text{: If } u \text{ is } A_1 \text{ and } v \text{ is } B_1, \text{ then } w \text{ is } C_1;$$
$$R_2\text{: If } u \text{ is } A_2 \text{ or } v \text{ is } B_2, \text{ then } w \text{ is } C_2.$$

*where $A_1 = \langle(0.2, 0.5, 0.8), (0.4, 0.5, 0.5), (0.9, 0.6, 0.3)\rangle$, $A_2 = \langle(0.1, 0.6, 0.7), (0.2, 0.4, 0.4), (0.9, 0.3, 0.3)\rangle$; $B_1 = \langle(0.1, 0.6, 0.9), (0.5, 0.6, 0.2), (0.8, 0.5, 0.1)\rangle$, $B_2 = \langle(0.3, 0.7, 0.7), (0.4, 0.5, 0.2), (0.8, 0.3, 0.2)\rangle$; $C_1 = \langle(0.2, 0.4, 0.7), (0.6, 0.5, 0.6), (0.9, 0.5, 0.2)\rangle$, $C_2 = \langle(0.2, 0.7, 0.8), (0.5, 0.5, 0.6), (0.7, 0.4, 0.3)\rangle$ are three-way fuzzy sets on U, V and W, respectively.*

*Given that the input is "u is $A^*$ and v is $B^*$", where $A^* = \langle(0.1, 0.6, 0.8), (0.3, 0.4, 0.2), (0.9, 0.5, 0.4)\rangle$, $B^* = \langle(0.2, 0.5, 0.7), (0.6, 0.5, 0.4), (0.8, 0.5, 0.3)\rangle$ are three-way fuzzy sets on U and V, respectively. Find the output w according to the triple I algorithm based on Zadeh's three-way fuzzy implication operator.*

*(1) Calculate the three-way fuzzy relation corresponding to each rule. The three-way fuzzy matrices $R_1$ and $R_2$ are used to represent the three-way fuzzy relations, which are as follows:*

$$R_1 = ((A_1)^T \wedge_t B_1) \to_z C_1 = \begin{bmatrix} (0.9, 0.4, 0.1) & (0.9, 0.4, 0.1) & (0.9, 0.4, 0.1) \\ (0.8, 0.5, 0.2) & (0.8, 0.5, 0.2) & (0.8, 0.5, 0.2) \\ (0.8, 0.5, 0.2) & (0.8, 0.5, 0.2) & (0.8, 0.5, 0.2) \\ (0.9, 0.4, 0.1) & (0.9, 0.4, 0.1) & (0.9, 0.4, 0.1) \\ (0.5, 0.5, 0.4) & (0.5, 0.5, 0.4) & (0.5, 0.5, 0.4) \\ (0.5, 0.5, 0.4) & (0.5, 0.5, 0.4) & (0.5, 0.5, 0.4) \\ (0.9, 0.4, 0.1) & (0.9, 0.4, 0.1) & (0.9, 0.4, 0.1) \\ (0.3, 0.0, 0.5) & (0.5, 0.0, 0.5) & (0.5, 1.0, 0.3) \\ (0.3, 0.0, 0.7) & (0.6, 0.5, 0.6) & (0.8, 1.0, 0.3) \end{bmatrix}.$$

$$R_2 = ((A_2)^T \vee_t B_2) \to_z C_2 = \begin{bmatrix} (0.7, 0.3, 0.3) & (0.7, 0.3, 0.3) & (0.7, 0.3, 0.3) \\ (0.2, 0.5, 0.4) & (0.4, 0.0, 0.4) & (0.4, 1.0, 0.3) \\ (0.2, 0.7, 0.8) & (0.5, 0.5, 0.6) & (0.7, 0.4, 0.3) \\ (0.4, 1.0, 0.3) & (0.4, 1.0, 0.3) & (0.4, 1.0, 0.3) \\ (0.2, 0.5, 0.4) & (0.4, 0.0, 0.4) & (0.4, 1.0, 0.3) \\ (0.2, 0.7, 0.8) & (0.5, 0.5, 0.6) & (0.7, 0.4, 0.3) \\ (0.3, 0.0, 0.8) & (0.5, 0.5, 0.6) & (0.7, 0.4, 0.3) \\ (0.2, 0.7, 0.8) & (0.5, 0.5, 0.6) & (0.7, 0.4, 0.3) \\ (0.2, 0.7, 0.8) & (0.5, 0.5, 0.6) & (0.7, 0.4, 0.3) \end{bmatrix}.$$

*(2) Calculate the input. Represent the input as the three-way fuzzy set on $U \times V$:*

$$A^* and B^* = (A^*)^T \wedge_t B^* = \begin{bmatrix} (0.1, 0.6, 0.8) & (0.1, 0.6, 0.8) & (0.1, 0.6, 0.8) \\ (0.2, 0.5, 0.7) & (0.3, 1.0, 0.4) & (0.3, 1.0, 0.3) \\ (0.2, 0.5, 0.7) & (0.6, 0.5, 0.4) & (0.8, 1.0, 0.4) \end{bmatrix}.$$

Since $((A^*)^T \wedge_t B^*)^{ct} <_t R$ and $R^{ct} <_t R$, the three-way fuzzy matrices $R_1$ and $R_2$ are converted to $R_{1*}$ and $R_{2*}$, which are as follows:

$$R_{1*} = \begin{bmatrix} (0.9, 0.4, 0.1) & (0.9, 0.4, 0.1) & (0.9, 0.4, 0.1) \\ (0.0, 0.0, 1.0) & (0.0, 0.0, 1.0) & (0.0, 0.0, 1.0) \\ (0.0, 0.0, 1.0) & (0.0, 0.0, 1.0) & (0.0, 0.0, 1.0) \\ (0.9, 0.4, 0.1) & (0.9, 0.4, 0.1) & (0.9, 0.4, 0.1) \\ (0.0, 0.0, 1.0) & (0.0, 0.0, 1.0) & (0.0, 0.0, 1.0) \\ (0.0, 0.0, 1.0) & (0.0, 0.0, 1.0) & (0.0, 0.0, 1.0) \\ (0.9, 0.4, 0.1) & (0.9, 0.4, 0.1) & (0.9, 0.4, 0.1) \\ (0.0, 0.0, 1.0) & (0.0, 0.0, 1.0) & (0.5, 1.0, 0.3) \\ (0.0, 0.0, 1.0) & (0.0, 0.0, 1.0) & (0.8, 1.0, 0.3) \end{bmatrix}.$$

$$R_{2*} = \begin{bmatrix} (0.0, 0.0, 1.0) & (0.0, 0.0, 1.0) & (0.0, 0.0, 1.0) \\ (0.0, 0.0, 1.0) & (0.0, 0.0, 1.0) & (0.0, 0.0, 1.0) \\ (0.0, 0.0, 1.0) & (0.0, 0.0, 1.0) & (0.0, 0.0, 1.0) \\ (0.0, 0.0, 1.0) & (0.0, 0.0, 1.0) & (0.0, 0.0, 1.0) \\ (0.0, 0.0, 1.0) & (0.0, 0.0, 1.0) & (0.4, 1.0, 0.3) \\ (0.0, 0.0, 1.0) & (0.0, 0.0, 1.0) & (0.7, 0.4, 0.3) \\ (0.0, 0.0, 1.0) & (0.0, 0.0, 1.0) & (0.0, 0.0, 1.0) \\ (0.0, 0.0, 1.0) & (0.0, 0.0, 1.0) & (0.7, 0.4, 0.3) \\ (0.0, 0.0, 1.0) & (0.0, 0.0, 1.0) & (0.7, 0.4, 0.3) \end{bmatrix}.$$

*(3) Calculate the inference result of each rule, respectively:*

$$C^{*1} = ((A^*)^T \wedge_t B^*) \circ R_{1*} = \langle (0.2, 0.5, 0.7), (0.2, 0.5, 0.7), (0.8, 1.0, 0.4) \rangle.$$

$$C^{*2} = ((A^*)^T \wedge_t B^*) \circ R_{2*} = \langle (0.0, 0.0, 0.1), (0.0, 0.0, 0.1), (0.7, 0.0, 0.3) \rangle.$$

*(4) Calculate the output. Perform a three-way fuzzy union on the inference results of the two rules to obtain the final result:*

$$C^* = C^{*1} \vee_t C^{*2} = \langle (0.2, 0.5, 0.7), (0.2, 0.5, 0.7), (0.8, 0.0, 0.3) \rangle. \tag{28}$$

*In order to illustrate the advantages of the triple I algorithm, the final result of the TCRI method is given as follows:*

$$C^* = \langle (0.3, 1.0, 0.4), (0.6, 0.0, 0.4), (0.8, 0.0, 0.3) \rangle. \tag{29}$$

*By comparing (28) and (29), it is obvious that* $\langle (0.2, 0.5, 0.7), (0.2, 0.5, 0.7), (0.8, 0.0, 0.3) \rangle \leq_t$ $\langle (0.3, 1.0, 0.4), (0.6, 0.0, 0.4), (0.8, 0.0, 0.3) \rangle$. *So, the triple I algorithm is better than the TCRI method.*

**Example 5.** *Let the universes $U$, $V$ and $V$, the three-way fuzzy sets $A_1$, $A_2$, $B_1$, $B_2$, $C_1$, $C_2$, and $A^*$, $B^*$ be the same as those in Example 4. Find the output w according to the triple I algorithm based on $R_0$ three-way residual implication.*

*(1) Calculate the three-way fuzzy relation corresponding to each rule. The three-way fuzzy matrices $R_1$ and $R_2$ are used to represent the three-way fuzzy relations, which are as follows:*

$$R_1 = ((A_1)^T \wedge_t B_1) \rightarrow_{r_0} C_1 = \begin{bmatrix} (1.0,1.0,0.0) & (1.0,1.0,0.0) & (1.0,1.0,0.0) \\ (1.0,1.0,0.0) & (1.0,1.0,0.0) & (1.0,1.0,0.0) \\ (1.0,1.0,0.0) & (1.0,1.0,0.0) & (1.0,1.0,0.0) \\ (1.0,1.0,0.0) & (1.0,1.0,0.0) & (1.0,1.0,0.0) \\ (0.5,0.5,0.4) & (0.6,0.0,0.4) & (1.0,1.0,0.0) \\ (0.5,0.5,0.4) & (0.6,0.0,0.4) & (1.0,1.0,0.0) \\ (1.0,1.0,0.0) & (1.0,1.0,0.0) & (1.0,1.0,0.0) \\ (0.3,0.0,0.5) & (0.6,0.0,0.5) & (1.0,1.0,0.0) \\ (0.3,0.0,0.7) & (0.6,0.5,0.6) & (1.0,1.0,0.0) \end{bmatrix}.$$

$$R_2 = ((A_2)^T \vee_t B_2) \rightarrow_{r_0} C_2 = \begin{bmatrix} (0.7,0.3,0.3) & (1.0,1.0,0.0) & (1.0,1.0,0.0) \\ (0.2,0.5,0.4) & (0.5,0.0,0.4) & (0.7,0.4,0.3) \\ (0.2,0.7,0.8) & (0.5,0.5,0.6) & (0.7,0.4,0.3) \\ (0.4,1.0,0.3) & (0.5,0.0,0.3) & (1.0,1.0,0.0) \\ (0.2,0.5,0.4) & (0.5,0.0,0.4) & (0.7,0.4,0.3) \\ (0.2,0.7,0.8) & (0.5,0.5,0.6) & (0.7,0.4,0.3) \\ (0.3,0.0,0.8) & (0.5,0.5,0.6) & (0.7,0.4,0.3) \\ (0.2,0.7,0.8) & (0.5,0.5,0.6) & (0.7,0.4,0.3) \\ (0.2,0.7,0.8) & (0.5,0.5,0.6) & (0.7,0.4,0.3) \end{bmatrix}.$$

*(2) Calculate the input. Represent the input as the three-way fuzzy set on $U \times V$:*

$$A^* \, and \, B^* = (A^*)^T \wedge_t B^* = \begin{bmatrix} (0.2,0.8,0.9) & (0.2,0.8,0.9) & (0.2,0.8,0.9) \\ (0.3,0.5,0.9) & (0.6,0.5,0.5) & (0.6,0.5,0.5) \\ (0.3,0.5,0.9) & (0.7,0.3,0.4) & (0.9,0.1,0.3) \end{bmatrix}.$$

*Since $((A^*)^T \wedge_t B^*)^{ct} <_t R$, the three-way fuzzy matrices $R_1$ and $R_2$ are converted to $R_{1*}$ and $R_{2*}$, which are as follows:*

$$R_{1*} = \begin{bmatrix} (1.0,1.0,0.0) & (1.0,1.0,0.0) & (1.0,1.0,0.0) \\ (1.0,1.0,0.0) & (1.0,1.0,0.0) & (1.0,1.0,0.0) \\ (1.0,1.0,0.0) & (1.0,1.0,0.0) & (1.0,1.0,0.0) \\ (1.0,1.0,0.0) & (1.0,1.0,0.0) & (1.0,1.0,0.0) \\ (0.0,0.0,1.0) & (0.0,0.0,1.0) & (1.0,1.0,0.0) \\ (0.0,0.0,1.0) & (0.0,0.0,1.0) & (1.0,1.0,0.0) \\ (1.0,1.0,0.0) & (1.0,1.0,0.0) & (1.0,1.0,0.0) \\ (0.0,0.0,1.0) & (0.6,0.0,0.5) & (1.0,1.0,0.0) \\ (0.0,0.0,1.0) & (0.6,0.5,0.6) & (1.0,1.0,0.0) \end{bmatrix}.$$

$$R_{2*} = \begin{bmatrix} (0.0,0.0,1.0) & (1.0,1.0,0.0) & (1.0,1.0,0.0) \\ (0.0,0.0,1.0) & (0.0,0.0,1.0) & (0.0,0.0,1.0) \\ (0.0,0.0,1.0) & (0.0,0.0,1.0) & (0.0,0.0,1.0) \\ (0.0,0.0,1.0) & (0.0,0.0,1.0) & (1.0,1.0,0.0) \\ (0.0,0.0,1.0) & (0.0,0.0,1.0) & (0.7,0.4,0.3) \\ (0.0,0.0,1.0) & (0.0,0.0,1.0) & (0.7,0.4,0.3) \\ (0.0,0.0,1.0) & (0.0,0.0,1.0) & (0.0,0.0,1.0) \\ (0.0,0.0,1.0) & (0.5,0.5,0.6) & (0.7,0.4,0.3) \\ (0.0,0.0,1.0) & (0.5,0.5,0.6) & (0.7,0.4,0.3) \end{bmatrix}.$$

*(3) Calculate the inference results of each rule, respectively:*

$$C^{*1} = (A^*)^T \wedge_t B^* \circ R_{1*} = \langle (0.2,0.5,0.7), (0.6,0.0,0.5), (0.8,0.0,0.3) \rangle.$$

$$C^{*2} = (A^*)^T \wedge_t B^*) \circ R_{2^*} = \langle (0.0, 0.0, 1.0), (0.5, 0.5, 0.6), (0.7, 0.0, 0.3) \rangle.$$

*(4) Calculate the output. Perform a three-way fuzzy union on the inference results of the two rules to obtain the final result:*

$$C^* = C^{*1} \vee_t C^{*2} = \langle (0.2, 0.5, 0.7), (0.6, 0.0, 0.5), (0.8, 0.0, 0.3) \rangle. \tag{30}$$

*In order to illustrate the advantages of the triple I algorithm, the final result of the TCRI method is given as follows:*

$$C^* = \langle (0.3, 1.0, 0.4), (0.6, 0.0, 0.4), (0.8, 0.0, 0.3) \rangle. \tag{31}$$

*By comparing (30) and (31), since* $0.2 < 0.3$, $0.7 > 0.4$, $(0.2, 0.5, 0.7) \leq_t (0.3, 1.0, 0.4)$; *since* $0.6 = 0.6$, $0.5 > 0.4$, $(0.6, 0.0, 0.5) \leq_t (0.6, 0.0, 0.4)$; $(0.8, 0.0, 0.3) = (0.8, 0.0, 0.3)$. *Thus,* $\langle (0.2, 0.5, 0.7), (0.6, 0.0, 0.5), (0.8, 0.0, 0.3) \rangle \leq_t \langle (0.3, 1.0, 0.4), (0.6, 0.0, 0.4), (0.8, 0.0, 0.3) \rangle$. *So, the result of the triple I algorithm is smaller than the TCRI method. That is, the same conclusion as in Example 4 is obtained.*

From the above two examples, it is clear that no matter which three-way fuzzy implication is used, the triple I algorithm is significantly better than the TCRI method.

## 5. Application of Three-Way Fuzzy Inference

Based on the above research on three-way fuzzy inference, the three-way fuzzy controller is constructed systematically in this section.

A Three-way fuzzy controller with dual input (error: $e$ and change rate of error: $ec$) and single output (control variable: $u$) is considered. Suppose that the basic universe of $e$, $ec$ and $u$ is the same, which is $[-6, 6]$, and their language value is represented by seven three-way fuzzy subsets: NB, NM, NS, ZE, PS, PM, PB.

(1) Three-way fuzzification. The trapezoidal, triangular truth-membership function, indeterminacy-membership function and falsity-membership function are used to realize the three-way fuzzification of $e$, $ec$ and $u$. The truth-membership functions of $e$, $ec$ and $u$ are: NB ($[-6, -4]$), NM ($[-6, -2]$), NS ($[-4, 0]$), ZE ($[-2, 2]$), PS ($[0, 4]$), PM ($[2, 6]$), PB ($[4, 6]$) as shown in Figure 1a, the indeterminacy-membership functions of $e$, $ec$ and $u$ are: NB ($[-6, -5]$), NM ($[-5, -3]$), NS ($[-3, -1]$), ZE ($[-1, 1]$), PS ($[1, 3]$), PM ($[3, 5]$), PB ($[5, 6]$) as shown in Figure 1b, and the falsity-membership functions of $e$, $ec$ and $u$ are: NB ($[-6, -3]$), NM ($[-6, -1]$), NS ($[-5, 1]$), ZE ($[-3, 3]$), PS ($[-1, 5]$), PM ($[1, 6]$), PB ($[3, 6]$) as shown in Figure 1c.

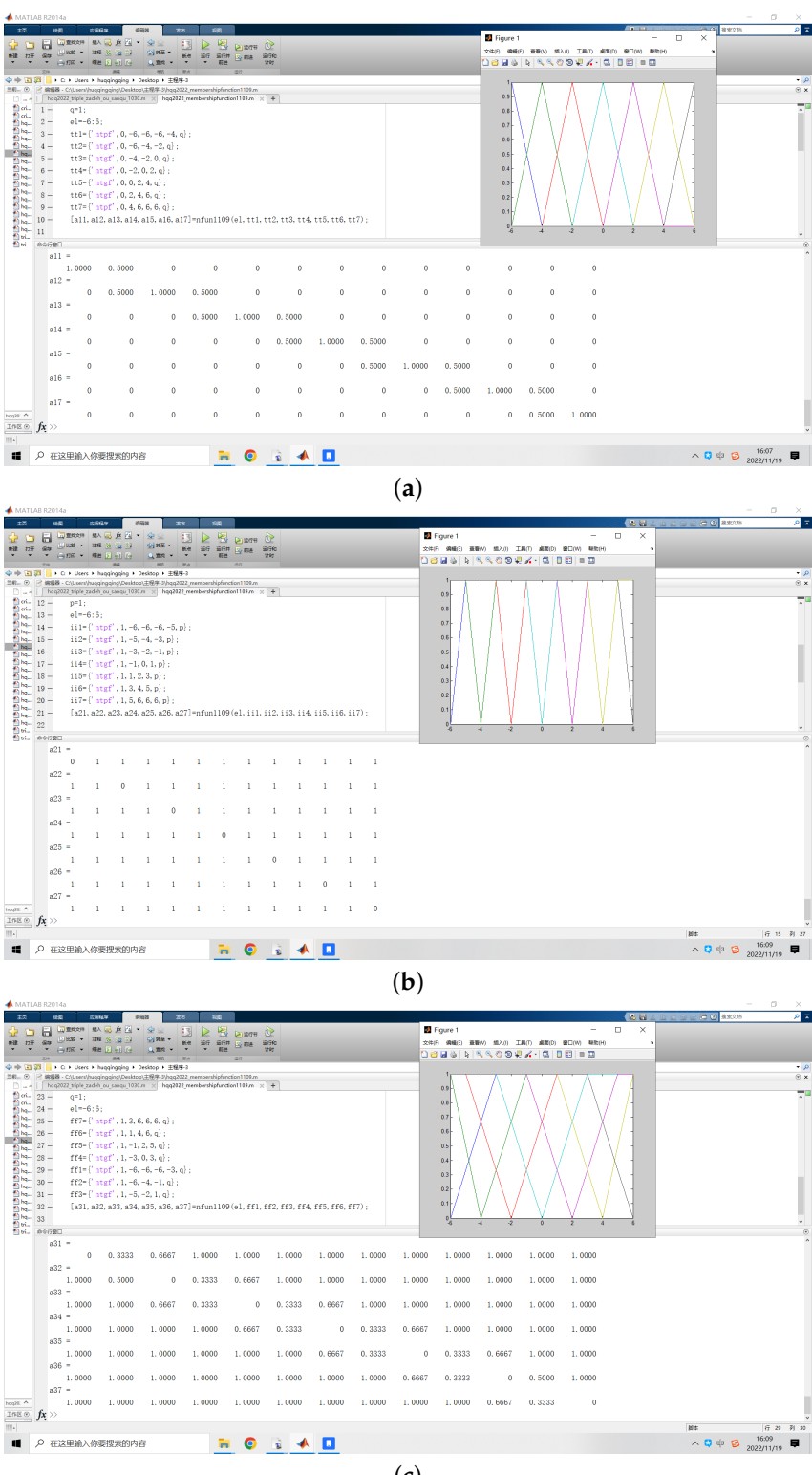

**Figure 1.** The truth-membership functions (**a**), indeterminacy-membership functions (**b**) and falsity-membership functions (**c**) of three-way fuzzy subsets of $e$, $ec$ and $u$.

(2) Three-way fuzzy control rules. The form of rules is "if $e$ is $A_i$ and $ec$ is $B_j$, then $u$ is $C_{ij}$", and the rules of fuzzy controller are used in the three-way fuzzy controller, which are shown in Table 1.

**Table 1.** Three-way fuzzy control rules.

| $C_{ij} \backslash B_j$ / $A_i$ | NB | NM | NS | ZE | PS | PM | PB |
|---|---|---|---|---|---|---|---|
| NB | PB | PB | PM | PM | PS | PS | ZE |
| NM | PB | PM | PM | PS | PS | ZE | NS |
| NS | PM | PM | PS | PS | ZE | NS | NS |
| ZE | PM | PS | PS | ZE | NS | NS | NM |
| PS | PS | PS | ZE | NS | NS | NM | NM |
| PM | PS | ZE | NS | NS | NM | NM | NB |
| PB | ZE | NS | NS | NM | NM | NB | NB |

where $A_i$, $B_j$, $C_{ij}(i, j = 1, 2, \ldots, 7)$ are three-way fuzzy subsets of $e$, $ec$, $u$, respectively.

(3) Three-way fuzzy inference. Triple I algorithms based on Zadeh's three-way fuzzy implication operator and $R_0$ three-way residual implication are applied to calculate the inference results of each rule, and then three-way fuzzy union is performed on the inference results of all rules to get the final result. The main MATLAB code of three-way fuzzy inference is shown in Figure 2.

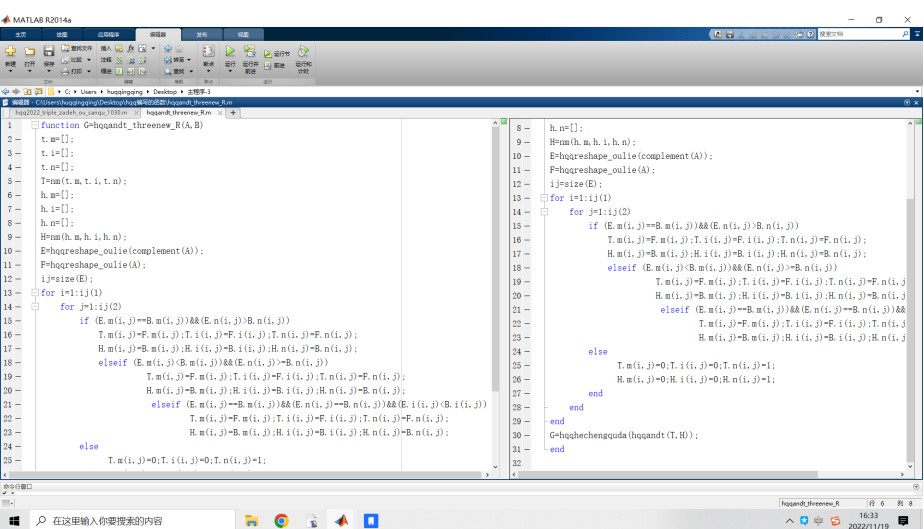

**Figure 2.** Three-way fuzzy inference based on the proposed triple I algorithm.

(4) Defuzzification. This paper mainly discusses the following two defuzzification methods :

(i) Mean of maximum membership method (abbreviated as MOM):

Let $U$ be a universe, $TFS1(U)$ be the ordinary three-way fuzzy sets on $U$. $B^*, (B_0)^* \in TFS1(U)$. Suppose $B^*(y_i) = \langle f_{B^*}(y_i), g_{B^*}(y_i), h_{B^*}(y_i) \rangle, \forall y_i \in U, i = 1, 2 \ldots k$. Then

$$(B_0)^*(y^*) = \langle f_{(B_0)^*}(y^*), g_{(B_0)^*}(y^*), h_{(B_0)^*}(y^*) \rangle = \vee_t \langle f_{B^*}(y_i), g_{B^*}(y_i), h_{B^*}(y_i) \rangle, y^* \in U.$$

Take $\langle f_{(B_0)^*}(y^*), g_{(B_0)^*}(y^*), h_{(B_0)^*}(y^*) \rangle$ into the corresponding membership function, and use the maximum average method to obtain the accurate value $y$.

(ii) Weighted average method (abbreviated as WA):

Let $U$ be a universe, $TFS1(U)$ be the ordinary three-way fuzzy sets on $U$. $B^*, (B_0)^* \in TFS1(U)$. Suppose $B^*(y_i) = \langle f_{B^*}(y_i), g_{B^*}(y_i), h_{B^*}(y_i) \rangle, \forall y_i \in U, i = 1, 2 \ldots k$. Then

$$y = (2 + f_{(B_0)^*}(y^*) - g_{(B_0)^*}(y^*) - h_{(B_0)^*}(y^*))/3, y^* \in U.$$

where $f_{(B_0)^*}(y^*) = (\sum_i^k y_i * f_{B^*}(y_i)) / \sum_i^k f_{B^*}(y_i)$, $g_{(B_0)^*}(y^*) = (\sum_i^k y_i * g_{B^*}(y_i)) / \sum_i^k g_{B^*}(y_i)$, $h_{(B_0)^*}(y^*) = (\sum_i^k y_i * h_{B^*}(y_i)) / \sum_i^k h_{B^*}(y_i)$.

The main MATLAB code of WA and MOM are shown in Figure 3, respectively. MOM is applied to defuzzification in this paper. The accuracy value of control variabal *u* are shown in Figure 4.

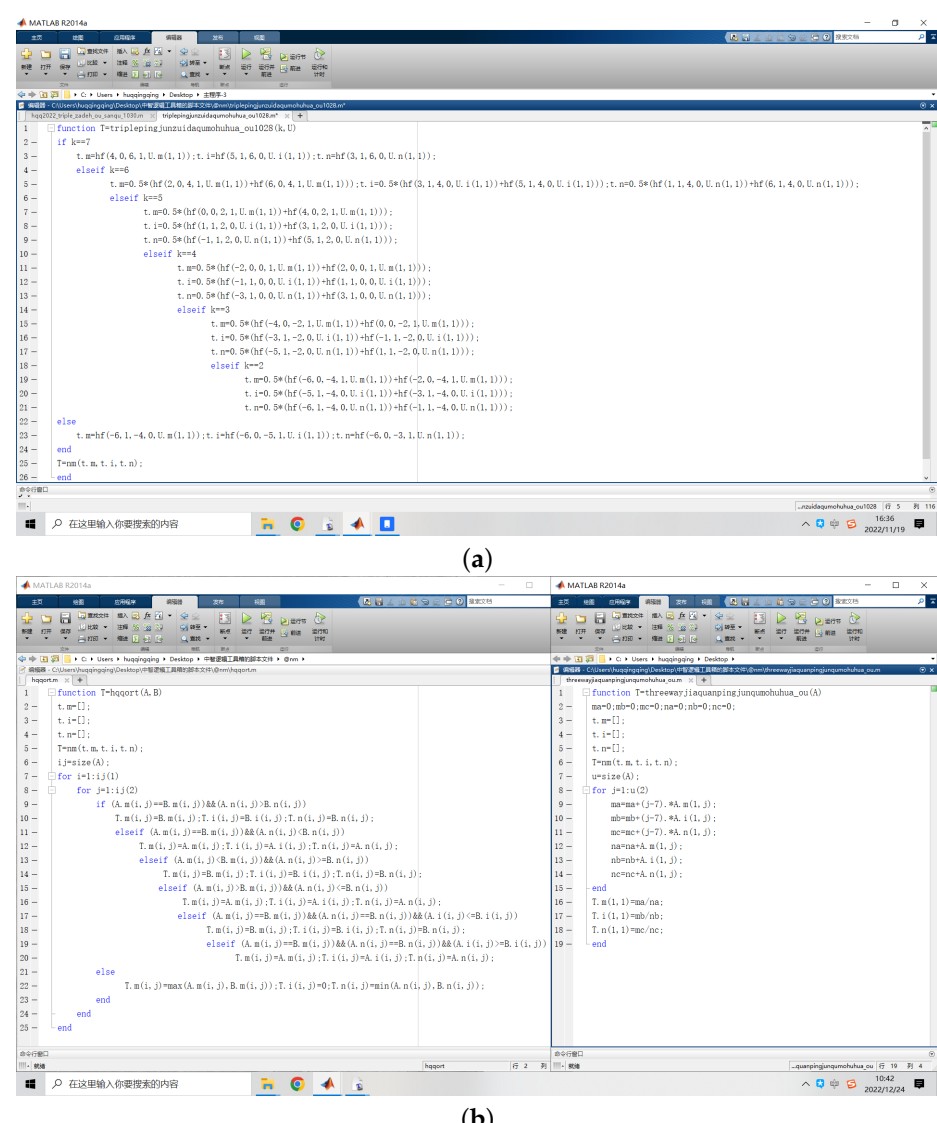

**Figure 3.** MOM defuzzification method (**a**) and WA defuzzification method (**b**).

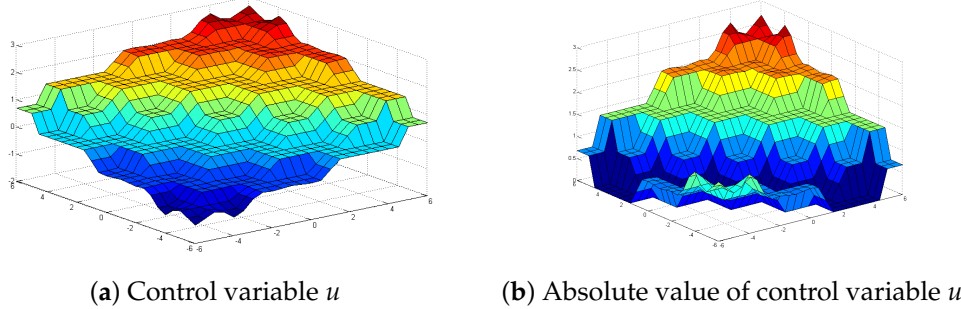

(**a**) Control variable *u*  (**b**) Absolute value of control variable *u*

**Figure 4.** Inference results of the triple I algorithm based on Zadeh's three-way fuzzy implication operator (or $R_0$ three-way residual implication).

Through experiments, we find that the inference result of the triple I method based on Zadeh's three-way fuzzy implication operator is the same as that of $R_0$ three-way residual implication, as described in Figure 4.

Based on different three-way fuzzy implication operators and different defuzzification methods, the three-dimensional surface diagrams of the control variables with large differences are obtained, which are shown in Table 2.

**Table 2.** Three-dimensional surface diagrams of control variable by different triple I algorithm.

| *u* ╲ IP ╲ DF | Mamdani | Gödel | Zadeh | $R_0$ |
|---|---|---|---|---|
| WA | Figure 4A [26] | Figure 5b | Figure 5b | Figure 5b |
| MOM | Figure 5a | Figure 4a | Figure 4a | Figure 4a |

where DF and IP are abbreviations of defuzzification and implication operator, respectively. $R_0$, Gödel three-way residual implications and Mamdani's and Zadeh's three-way fuzzy implication operator are considered.

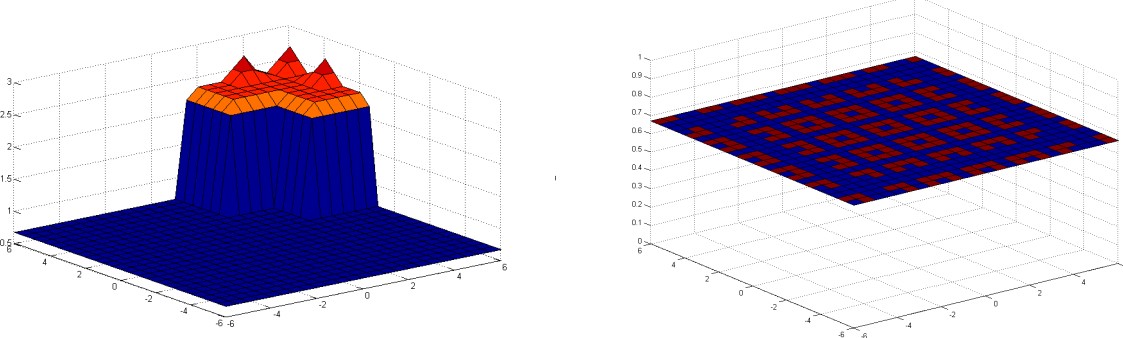

(**a**) Control variable *u* (Mamdani; MOM)    (**b**) Control variable *u* (Gödel, Zadeh and $R_0$; WA)

**Figure 5.** Inference results of triple I algorithm.

By comparing and analyzing the experimental results, we can find:

(1) According to the experimental results, When the defuzzification method (MOM or WA) is selected, no matter which implication operator is selected, the inference result of the triple I algorithm is the same. This shows that the defuzzification method has a great impact on the three fuzzy inference results.

(2) In the three-way fuzzy controller, if we choose the MOM defuzzification method and the triple I algorithm, which is based on Zadeh's three-way fuzzy implication operators and $R_0$, Gödel three-way residual implications, then the three-dimensional surface diagram of the control variable is not only the same but also better than that of Mamdani's three-way fuzzy implication operator. However, if we choose the WA defuzzification method, then it is much better to choose Mamdani's three-way fuzzy implication operator.

Therefore, for practical problems, it is particularly important to choose appropriate three-way fuzzy implication operators and defuzzification methods.

## 6. Conclusions

This paper mainly studies three-way fuzzy inference and its basic and important application in the design of a three-way fuzzy controller, which lays an important theoretical foundation for the construction of a three-way fuzzy control system. The research results of this paper mainly include the following aspects:

(1) From the perspective of three-way fuzzy logic, the proposed triple I algorithm is better than the existing TCRI method, because each step of three-way fuzzy inference has a strictly logical basis, so the inference process is more rigorous, and the inference results are more reasonable.

(2) By comparing the results of specific examples with the results of existing TCRI methods, it is proved that the proposed triple I algorithm can effectively improve TCRI methods and better reflect the internal relationship between three-way fuzzy inference and three-way fuzzy logic. Furthermore, it makes full theoretical preparation and algorithm implementation preparation for further research on the three-way fuzzy control based on the triple I algorithm.

(3) For the design of the three-way fuzzy controller, the three-dimensional surface diagram of the control variable obtained by the MOM method is better.

For the three-way fuzzy controller, the parameters of the truth-membership function, indeterminacy membership function and falsity-membership function also directly affect the three-way fuzzy inference result, so we will study the optimization method of three-way fuzzy controller. Furthermore, we will apply the triple I algorithm to solve the practical problems in industrial process control and realize the advantages of the three-way fuzzy inference methods in solving practical problems through simulation experiments with the actual industrial process data as the carrier. In addition, the applications of three-way fuzzy sets in image processing, neuro-fuzzy control uncertain decision-making and related algebraic systems [59–63] will be further studied.

**Author Contributions:** This paper was written through contributions of all authors. The individual contributions and responsibilities of all authors can be described as follows: The idea of this paper was put forward by X.Z. and Q.H. completed the existing research on three-way fuzzy inference and wrote the paper. All authors have read and agreed to the published version of the manuscript.

**Funding:** This work is supported by the National Natural Science Foundation of China under Grant Nos. 61976130 and 12201373.

**Institutional Review Board Statement:** Not applicable.

**Informed Consent Statement:** Not applicable.

**Data Availability Statement:** No data.

**Conflicts of Interest:** The authors declare no conflict of interest.

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
