# Peer review of "Three-Way Fuzzy Sets and Their Applications (III)"

_axioms, doi:10.3390/axioms12010057_

Round 1
Reviewer 1 Report
This paper studies fuzzy inference on Three-way Fuzzy Sets (TFSs). In particular, the authors adapt the well-known triple I algorithm to the case of TFSs, considering adapted fuzzy implications, and compare it with the previous Three-way Compositional Rule of Inference (abbreviated as TCRI).
Firstly, I think that this paper lacks a good motivation. It is not clear why this is an important research problem to be addressed, and why the results could be useful. In terms of practical applications, what real-world problems would benefit from this reasoning algorithm? Examples 2-5 and Section 5 are not useful for that.
Going even further, I have doubts about the practical usefulness of Three-way Fuzzy Sets. I read [24] and tried to read [23], but I got a "file not found" error. It is not clear to me why "three" and not any other number. I found some examples where there is a triple that could justify having a Three-way Fuzzy Sets such as Positive-Negative-Boundary or Symbols-Meaning-Value. Yes, in [24, Example 1], the triple is Mathematics-Sports-English. It seems clear that there could be less than 2 subjects or more than 3. So, again, why 3?
Even if we accept 3 as the magic number, what is the usefulness of TFS? Which problems can't be solved with standard fuzzy sets and can be solved with TFSs?
Then, in the current paper, the authors build the fuzzy inference on TFSs but, again, which problems can't be solved with standard fuzzy inference or are improved with this inference? In Section 5 ("Application of three-way fuzzy inference") the authors seems to solve a standard fuzzy control problem.
Furthermore, the whole approach relies on the order relation in Definition 2. However, such relation is not justified. x <= y implies x1 < y1 or x1 = y1 but, in practical applications, is this justified? Going back to [24, Example 1], this means that the "value" in Mathematics is smaller or equal than the value of Sports. Or, in [24, Example 2], that the red component of a pixel is smaller or equal than the green component. What is the rationale for that?
The paper is also organized in a confusing way:
- The proof of Theorem 1 should not be in the Preliminaries section, it should be in [23] (or, in a different part of this paper, maybe in an appendix).
- Similarly, if Theorem 2 is not original, it should not be in the Preliminaries.
- Definition 2 uses D*, but then (from Definition 4) the authors use D.
- Mamdani is not an implication, as mentioned by the authors in Remark 1. Therefore, it should not appear in Example 1 as a fuzzy implication, and in should not appear as a fuzzy implication in Section 4.2. The name of Section 4.2 could replace "fuzzy implication" with "Zadeh three-way or Mamdani operator".
- In 4.3, "Based on the above research" is confusing. What specific properties or results do you mean?
The paper is not convincing at showing that the proposed triple I algorithm can improve fuzzy reasoning.
- In Example 2, the authors mention that "in terms of the meaning of seeking the smallest B* on V, the triple I algorithm is better", as it provides a smaller value. However, in Examples 3 and 4, triple I algorithm provides a higher value. I cannot agree with the claim that "it is clear that no matter which three-way fuzzy implication is used, the triple I algorithm is significantly better than the TCRI method": it could be true but is not clear at all. Therefore, authors must explain more deeply why it is better in each example.
- Indeed, in Section 5, in the common case, "the output of the three-way fuzzy controller based on TCRI method is the same as that of triple I algorithm". So, with the same output, how can it effectively improve TCRI?
Some technical problems:
- Definitions 4-6 (implication, residuated implication, t-norm) are trivial, it suffices to use the order relation in Definition 2.
- Defuzzification operators are not defined on D([0, 1])
- In standard fuzzy logic, residual implications only exist if the t-norm is left-continuous. Something similar should happen here.
For all these reasons, I think that the paper is not ready to be published at its current state.
Minor comments:
- p. 6. "that satisfyings" -> "satisfying" or "that satisfies"
- p. 7. "Pones" -> "Ponens"
- Figures 1-4 are too small to be read.
Author Response
Dear Reviewer1,
Thank you very much for your good suggestions, we have studied comments carefully and have made correction which we hope meet with approval. Please see the attachment.
Sincerely yours, Qingqing Hu, Xiaohong Zhang

Reviewer 2 Report
In this paper, the authors introduced the three-way t-norm and its residual implication on D([0,1]), in order to propose the triple I algorithm based on three-way residual implication or based on a three-way fuzzy implication operator. They also proposed the three-way fuzzy inference based on the triple I algorithm and utilized it for an application. The results indicated the superiority of the triple I algorithm over the TCRI method. The paper is well-structured and provides interesting results. I propose to be accepted after minor revision because Figures 1-4 have to be presented more clearly since they are not readable.
Author Response
Dear Reviewer2,
Thank you very much for your good suggestions, we have studied comments carefully and have made correction which we hope meet with approval. Please see the attachment.
Sincerely yours,
Qingqing Hu, Xiaohong Zhang

Reviewer 3 Report
The paper contains some new ideas, but it also contains many incorrect statements.
1.1. In practice, the idea of the Three-way Fuzzy Sets (TFS) is a generalization of the Picture Fuzzy Sets (PFS), but this fact is not discussed anywhere. On the other hand, as discussed in the book:
Vassilev, P., K. Atanassov. Extensions and Modifications of Intuitionistic Fuzzy Sets. “Prof. Marin Drinov” Academic Publishing House, Sofia, 2019,
first, the PFSs are a partial case of the Interval-Valued Intuitionistic Fuzzy Sets (IVIFSs) , and second, some years before the introduction of the PFSs, they were introduced in:
Hinde, C., R. Patching. Inconsistent intuitionistic fuzzy sets. Developments in Fuzzy Sets, Intuitionistic Fuzzy Sets, Generalized Nets and Related Topics, Vol. 1, 2008, 133-153.
In addition, reference [12] is not the origin of the PFSs. They were introduced a year earlier in
Cuong, B. C., V. Kreinovich. Picture fuzzy sets-a new concept for computational intelligence problems, In: Proceedings of the Third World Congress on Information and Communication Technologies WICT'2013, Hanoi, Vietnam, December 15-18, 2013, 1-6.
On the other hand, in this form of the definition of the TFS, this notion corresponds to the definition of the Neutrosophic Set that is only cited without any additional discussion.
22. The expression
“x ≤∗ y if and only if x ≤ y, for every x, y ∈ L;”
is unclear.
33. It is not clear, why Theorem 1 is proved here, if it is proved in [1], but it raises the question: what is the relation between the TFSs and the Intuitionistic L-Fuzzy Sets? This question is not discussed in the paper, while it is of great importance if the authors pretend that their research represents a novelty.
44. The implications given in Example 1 have existed for many years BEFORE [23], but this is not mentioned. In addition, in the frames of intuitionistic fuzzy logic, these implications have explicit forms. Examples 2 – 5 do not convince the necessity of the use of TFSs.
55. A big part of the definitions and assertions from Sections 3 and 4 are analogues of well-known definitions and assertions, but without any citations of the original sources. By all these remarks, I cannot recommend publishing of the paper in the present form.
Author Response
Dear Reviewer 3,
Thank you very much for your good suggestions, we have studied comments carefully and have made correction which we hope meet with approval. Please see the attachment.
Sincerely yours,
Qingqing Hu, Xiaohong Zhang

Round 2
Reviewer 1 Report
This paper studies fuzzy inference on Three-way Fuzzy Sets (TFSs). In particular, the authors adapt the well-known triple I algorithm to the case of TFSs, considering adapted fuzzy implications, and compare it with the previous Three-way Compositional Rule of Inference (abbreviated as TCRI).
I would like to thank the authors for their response and for the improved version of the manuscript.
As I mentioned in my previous review, the whole approach relies on the order relation in Definition 2. While the authors tried to explain the rationale in the response to the reviewer, and I admit that my examples in the previous review were wrong, I still think that the order relation is not convincing. Following Example 1 in [24], given x = ⟨95.6, Good, 9⟩ and w = ⟨99, Good, 8.5⟩, is x <= w. That is, x <= w because the score in Mathematics of x is lower than the score of w, and the score in English of x is higher than the score of w. But, why Mathematics are more important than English? And why the score of English of x has to be higher? One would expect that if this score is lower, then this increases our confidence in x being less or equal than w. Following Example 2 in [24], if x has a lower red component and a higher blue component than w, then x <= w. Again, it is not clear why "red" is more important than "blue", and why x needs to have a higher value of blue.
Furthermore, some of the answers of the authors did not have any change in the text; in particular, the authors should explain shortly in the text why in Examples 3 and 4, triple I algorithm provides a lower value than the TCRI method.
Author Response
Dear Reviewer,
Thank you very much for your good suggestions, we have studied comments carefully and have made correction which we hope meet with approval. Please see the attachment.
Sincerely yours,
Qingqing Hu, Xiaohong Zhang

Reviewer 3 Report
Now, the paper is considerably better. My only remark is about the second change on page 3 (the lines are not numbered): there is a mistype.
Author Response

(The authors gave the same response as above.)

Round 3
Reviewer 1 Report
This paper studies fuzzy inference on Three-way Fuzzy Sets (TFSs). In particular, the authors adapt the well-known triple I algorithm to the case of TFSs, considering adapted fuzzy implications, and compare it with the previous Three-way Compositional Rule of Inference (abbreviated as TCRI).
I would like to thank the authors for their responses and for the effort to prepare a improved version of the manuscript. I think that this versions is acceptable for publication.